# L-Tryptophan-Dependent Auxin-Producing Plant-Growth-Promoting Bacteria Improve Seed Yield and Quality of Carrot by Altering the Umbel Order

Anam Noor [1], Khurram Ziaf [2,*], Muhammad Naveed [3,*], Khuram Shehzad Khan [3,4], Muhammad Awais Ghani [2], Iftikhar Ahmad [2], Raheel Anwar [2], Manzer H. Siddiqui [5], Amir Shakeel [6] and Azeem Iqbal Khan [6]

1. Department of Horticulture, Bahauddin Zakariya University, Multan 60800, Pakistan; anamnoor83@gmail.com
2. Institute of Horticultural Sciences, University of Agriculture, Faisalabad 38040, Pakistan; awais.ghani@uaf.edu.pk (M.A.G.); iftikharahmadhashmi@gmail.com (I.A.); raheelanwar@uaf.edu.pk (R.A.)
3. Institute of Soil and Environmental Sciences, University of Agriculture, Faisalabad 38040, Pakistan; khurramshehzadkhanniazi@gmail.com
4. College of Resources and Environmental Sciences, China Agricultural University, Beijing 100193, China
5. Department of Botany and Microbiology, College of Science, King Saud University, Riyadh 11451, Saudi Arabia; mhsiddiqui@ksu.edu.sa
6. Department of Plant Breeding and Genetics, University of Agriculture, Faisalabad 38040, Pakistan; dramirpbg@gmail.com (A.S.); ranaazeemiqbal@yahoo.com (A.I.K.)
* Correspondence: khurramziaf@uaf.edu.pk (K.Z.); muhammad.naveed@uaf.edu.pk (M.N.)

**Abstract:** Carrot (*Daucus carota* L.) seed quality is affected by umbel position due to uneven maturation of carrot seeds produced in different umbel orders. However, keeping this in view, we tested whether seed quality could be improved with the suppression of tertiary umbels under exogenous auxin application. Using auxin-producing bacterial isolates, i.e., *Bacillus* sp. MN54, *Enterobacter* sp. MN17, *Pantoea* sp. MN34, and *Burkholderia phytofirmans* PsJN, the arrangements of carrot umbel order were evaluated in terms of quality carrot seed production. The results revealed that auxin production by plant-growth-promoting rhizobacteria showed significant differences among measured growth indices, yield, and seed quality attributes. The selected endophytic strains co-applied with auxin via foliar application improved all growth- and yield-related traits, as well as the enzymatic activities of carrots. Noticeably, MN17+L-tryptophan and MN34+L-tryptophan effectively minimized the number of tertiary umbels by increasing the number of secondary umbels. Furthermore, treating with PsJN+L-tryptophan and MN34+L-tryptophan resulted in reduced conductivity of seed leachates and malondialdehyde levels in primary, secondary, and tertiary umbel seeds. These findings collectively indicate the potential of the foliar application of PsJN+L-tryptophan and MN34+L-tryptophan to effectively alter umbel arrangement, leading to improved yield and seed quality. This study implies that carrot seed producers can consider employing specific PGPB strains, particularly MN34+L-tryptophan, to suppress tertiary umbels and achieve higher yields of high-quality carrot seeds.

**Keywords:** *Enterobacter* sp. MN17; *B. phytofirmans* PsJN; *Pantoea* sp. MN34; *Bacillus* sp. MN54; *Daucus carota*; L-tryptophan; PGBP; seed quality

## 1. Introduction

Carrot (*Daucus carota* L.), from the family *Apiaceae*, is a winter vegetable crop with high demand [1]. It is a rich source of carotenoids, anthocyanins, dietary fiber, essential vitamins, and minerals [2]. Root color and flavor are the most important qualitative factors in varietal selection [3,4]. The origin of this crop is considered to be Central Asia and is now produced globally with medicinal and nutritional values [1,5]. The worldwide production of carrot is increasing, which is estimated to be about 27.39 million tonnes per year, in an

area of 9, 90 thousand hectares [6]. In Pakistan, carrot ranks third among winter vegetables and was cultivated in an area of 14.32 thousand hectares, with an annual production of 241.91 thousand tonnes, during the years 2017–2018 [7].

The anticipated demand for vegetable seeds in Pakistan stands at 5070 MT. However, merely 82 MT is accessible domestically, failing to meet the growers' need for seeds [8]. Due to the production of low-quality carrot seeds produced at a small level in Pakistan, a large quantity of the seeds available in the market is imported from other countries, due to which farmers have to pay high costs for the seeds [9,10]. Thus, to improve the production of carrot at the local level, the accessibility of quality seeds is crucial [11]. The yield and quality of carrot seeds can be improved by adopting improved practices, from seed planting to harvesting and postharvest treatment of seeds [10].

Umbel position (primary, secondary, or tertiary umbels) and size also influence the production of good-quality carrot seeds, e.g., vigor and germination [12,13]. Primary and secondary umbel seeds are of higher quality than tertiary umbel seeds [14]. Seed quality differs among various umbel orders because the seed-setting process begins with primary umbels, followed by the flowering of secondary umbels, while tertiary umbels are still emerging. Consequently, seeds in tertiary umbels mature much later, coinciding with the shattering of seeds from primary umbels [15–17], particularly in hot subtropical environments [18]. Carrot seed quality can be improved by altering the umbel order, i.e., the suppression of tertiary umbels [13]. Recently, the exogenous application of auxins has been reported not only to reduce tertiary umbels in carrot but also to improve the quality of seeds produced on auxin-treated plants [18].

Sustainable agriculture is fundamentally important because it offers the potential to meet future agricultural needs of fast-growing populations, which is beyond the capabilities of traditional agriculture [19]. The utilization of plant-growth-promoting bacteria, either rhizosphere bacteria or endophytes, is gaining popularity in various parts of the world [20,21]. Plant-growth-promoting rhizobacteria (PGPR) are beneficial bacteria that colonize plant roots [22] and various kinds of bacteria, such as *Pseudomonas*, *Azospirillum*, *Azotobacter*, *Klebsiella*, *Enterobacter* [23,24], *Alcaligenes*, *Arthrobacter*, *Burkholderia*, *Bacillus* [25], and *Serratia* [26], lead to increased plant growth via different mechanisms like phosphate solubilization [27], biological nitrogen fixation (BNF), siderophore production [28,29], and the production of phytohormones such as gibberellins, cytokinins and indole-3-acetic acid (IAA) [30]. PGPR can also modulate phytohormone levels in plant tissues [31].

Auxins play an important role in the growth and developmental processes of plants, such as cell division and enlargement, apical dominance control, flowering, fruiting, photosynthetic capacity, and the effective translocation of assimilates [32,33]. Tryptophan is a remarkable amino acid commonly found in root exudates. It is the main precursor molecule for the biosynthesis of indole-3-acetic acid in bacteria and higher plants [34]. Indole-3-acetic acid (IAA) is one of the most extensively studied and abundant types of auxins in plants. The production of IAA is affected by variations in different rhizobacteria species and strains, culture conditions, growth stages, and substrate availability [35]. The endogenous hormonal level of the plants can be altered with the exogenous application of phytohormones or their precursors [36].

It has been established that the biosynthesis of IAA by plant-associated bacteria with L-tryptophan (L-TRP) substrate improves the growth and yield of various crops [37–40]. Bacterial phytohormone production by rhizobacteria using L-tryptophan (L-TRP) as a stimulant from root exudates is responsible for increasing auxin production, which enhances root hair density and improves seed germination and plant growth [41]. Similarly, Khan et al. [42] stated that the endophyte *B. subtilis* LK14 improved shoot and root biomass and chlorophyll (a and b) contents in tomato by producing the highest amount of IAA. Previously, Noor et al. [18] reported that the exogenous application of synthetic auxins can potentially alter the umbel order in carrot via apical dominance control. However, the role of microbial auxin in controlling apical dominance has not been studied with regard to seed production in carrot.

Primarily, the objective of this study was to assess the effectiveness of bacterial strains in changing umbel order to enhance the production of high-quality carrot seeds. While numerous researchers have explored carrot root crops extensively, there has been no prior research published on the manipulation of umbel order through microbial application in carrot seed crops. More specifically, the present study is designed to evaluate the potential of plant-growth-promoting bacteria and L-tryptophan (an auxin precursor) for the growth enhancement and quality seed production of carrot under natural field conditions.

## 2. Materials and Methods

### 2.1. Plant Material and Experimental Site

The present experiments were conducted at the Vegetable Experimental Area (Latitude 31°31′ N, Longitude 73°10′ E, and altitude 213 m) and Vegetable Seed Lab, Institute of Horticultural Sciences, University of Agriculture, Faisalabad, Pakistan. The seeds of locally available carrot cv. T-29 were planted. The same experiment was repeated for two years to evaluate the best PGPR strain (s) for altering umbel order and improving carrot seed yield and quality.

### 2.2. Experimental Details and Treatments Plan

Seeds of carrot cv. T-29 were sown during September in both years, on both sides of raised beds prepared 75 cm apart. First irrigation was applied just after seed sowing, while subsequent irrigations were applied as per crop requirement. Recommended fertilizers (125:125:123 kg ha$^{-1}$) were applied using urea in three splits as a nitrogen source, while diammonium phosphate (DAP) and sulfate of potash (SOP) were used before ridge preparation as a source of phosphorous and potash, respectively. Carrots were ready for steckling preparation after 120 days of sowing. Off-type (with poor color) and small-sized roots, as well as, cracked, injured, diseased, and forked roots, were discarded, and healthy standard-sized roots (25–30 cm in length and 7–9 cm in diameter) were selected to prepare stecklings for seed production. The stecklings were prepared by removing the lower one-third portion of the root and also keeping leaf bases (5 cm) intact. To perform the stecklings' plantation, the soil was prepared to a suitable texture, and prior to forming ridges, a portion of nitrogen, along with the complete amounts of phosphorus and potash, were administered. The remaining nitrogen, however, was divided into two separate applications [18]. Uniform-sized stecklings were planted at the top of ridges, which were 75 cm apart, keeping a distance of 30 cm between stecklings, during the first week of January of both years. The field was irrigated after steckling plantation, and the pre-emergence herbicide Stomp Xtra (FMC Limited) was sprayed within 24 h of irrigation. Moreover, a randomized complete block design (RCBD) experiment was conducted with treatments comprising different PGPB strains with or without auxin precursor L-TRP: Control (distilled water), L-TRP (10$^{-4}$ M), MN54, MN54+L-TRP, MN17, MN17+L-TRP, MN34, MN34+L-TRP, PsJN, and PsJN+L-TRP. Each treatment was applied three times at 15-day intervals, starting from steckling's planting time. During both years in the field, each treatment was replicated three times (each replicate comprised 10 plants) while four replicates were used in seed quality assessment (see Supplementary Figure S1).

Auxin precursor (L-TRP) and/or PGPB strains (*Bacillus* sp. MN54, *Enterobacter* sp. MN17, *Pantoea* sp. MN34, and *Burkholderia phytofirmans* PsJN) were used for foliar application. Bacterial culture (300 mL per plant) was sprayed, and distilled water (DW) was sprayed as a control using a separate sprayer pump for each treatment. Carrot seeds were harvested after 120 days of planting the stecklings in May of each year. The mature umbels were harvested when the umbels attained tan color. Seeds were then threshed by hand rubbing, cleaned, and dried until they achieved safe moisture (7–8%) level. Seed lots were stored under ambient laboratory conditions.

## 2.3. Preparation of Bacterial Inoculum

The inoculants of selected bacterial strains (MN54, MN17, MN34, and PsJN) were prepared in Erlenmeyer flasks containing 10% tryptic soya broth (TSB) and incubated at $28 \pm 2\ °C$ for 48 h in the orbital shaking incubator (Firstek Scientific, Tokyo, Japan) at 180 rev min$^{-1}$. The optical density of the broth was adjusted to 0.5 measured at 600 nm using a spectrophotometer (Evolution 300 LC, Cambridge, UK) to obtain a uniform population of bacteria ($10^8$–$10^9$ colony-forming units (CFU ml$^{-1}$) in the broth at the time of inoculation [43].

## 2.4. Data Collection and Measurements

Data on vegetative, floral, and yield traits were measured in the field. Seed quality and biochemical attributes of seeds from umbels of various orders were studied. Carrot stecklings were uprooted after 45 days of sowing and the fresh weight of the adventitious root of five randomly selected stecklings was determined. Similarly, the height of five randomly selected plants was recorded from the base to the apex of the primary umbel at the flowering stage. All secondary and tertiary umbels of ten randomly selected plants were counted after harvesting and calculated.

## 2.5. Seed Quality Assessment

Seeds from all primary, secondary, and tertiary umbels of a plant were extracted separately, winnowed to remove inert matter, and weighed. The harvested seeds from different umbel orders were subjected to a germination test, performed for 7 days at $24 \pm 2\ °C$, with four replicates of 25 seeds each following rules of the Association of Official Seed Analysts [44]. Data were recorded for seed germination (% age) and vigor index [45]. The electrical conductivity of seed leachates (an indicator of membrane damage) was determined after 24 h by soaking seeds (1.0 g) in 50 mL of deionized water, using a digital conductivity meter (Cyberscan Con 11, Eutech Instruments, Singapore). The seed conductivity of primary, secondary, and tertiary umbel was determined. To estimate lipid peroxidation in carrot seeds, the malondialdehyde contents were assayed according to Heath and Packer [46]. For the estimation of antioxidant enzyme activities, samples (1 g) were homogenized in 2 mL of phosphate buffer (pH 7.2) in a prechilled mortar and pestle. After thorough centrifugation at 10,000 g for 10 min, in a microcentrifuge (235-A, Pegasus Scientific Inc., Frederick, MD, USA), the supernatant was collected in Eppendorf tubes and used for further assays of several enzymes. The activity of catalase (CAT) (EC 1.11.1.6) and peroxidase (POD) (EC 1.11.1.7) enzymes were analyzed according to Liu et al. [47]. Superoxide (SOD) dismutase (EC 1.15.1.1) activity was assessed via 50% inhibition of photochemical reduction of nitro-blue-tetrazolium (NBT) as outlined by Stajner and Popovic [48]. The total antioxidant contents of carrot seeds were determined as described by Razzaq et al. [49] using a DPPH assay. The total phenolic contents of the harvested carrot seeds were measured according to Ainsworth and Gillespie [50].

## 2.6. Colonization of Selected Endophytic Bacteria from Rhizosphere, Root, and Shoot

The persistence of the selected bacterial strains was assessed through dilution and the plate-counting method. For the colonization test, rhizosphere soil was sampled, and soil slurry was established (1:5 soil: NaCl 0.9%) following incubation for 30 min at $28 \pm 1\ °C$. After the sediments settled down, sequential dilutions of about $10^{-6}$ were placed on a 10% TSB medium. Colonies were calculated after preserving the Petri dishes at room temperature for two days, and colonization reading was performed using the following formula [21]:

$$\text{No of colonies (CFU g}^{-1}\text{ dry weight)} = \frac{\frac{1}{\text{dilution factor}} \times \text{no of colonies } (10^{-5})}{\text{dry weight}}$$

where CFU represents the colony-forming unit.

### 2.7. Measurement of the Auxin Produced by Bacterial Strains

The amount of auxin produced by *B. phytofirmans* PsJN and other strains, i.e., MN17, MN34, and MN54, was also calculated using a colorimetric method in the form of IAA equivalents [51]. The bacterial cultures developed in LB broth along with 1% L-TRP solution were extracted via centrifugation. The supernatants of 3 mL were mixed with 2 mL 12 g L$^{-1}$ of FeCl$_3$ in 429 mL L$^{-1}$ of H$_2$SO$_4$ (Salkowski's reagent). Afterward, homogenous mixtures were left at ambient temperature for 30 min until color development and optical density were recorded at 535 nm with a spectrophotometer. The amount of auxin produced by bacterial strains was determined using a standard curve for IAA prepared from dilutions in the range of 10–100 μg mL$^{-1}$.

### 2.8. Statistical Analysis

Statistical software (Statistix 8.1) was used to analyze the study indices over the two years using the ANOVA technique [52]. Treatments' mean values were computed for significant differences using Tukey's test ($p \leq 0.05$).

### 3. Results

Plant height showed significant differences for foliar treatments during both years (Table 1). Among foliar treatments, maximum plant height (127.2 and 118.2 cm) was recorded for MN54+L-TRP during both years, i.e., a 30% increment in plant height. The minimum height of the plant (94.4 and 93.8 cm) was observed for PsJN, but it was not significantly different from PsJN+L-TRP for Year I (Table 1). Adventitious root weight was significantly affected by the foliar treatments. The highest weight of adventitious root (1.56 and 1.82 g) on the 30th day after steckling plantation was recorded for MN34+L-TRP during both years. Moreover, the weight of adventitious roots was 42% higher than that of untreated plants (Table 1). The maximum adventitious root weight (1.96 and 2.50 g) on the 45th day after steckling plantation was observed in plants treated with MN34+L-TRP during Years I and II, respectively, with 45% more root weight recorded for MN34+L-TRP than control (Table 1).

**Table 1.** The effects of foliar treatments on plant height and adventitious root weight of stecklings of carrot cv. T-29.

| Treatments | | | | |
|---|---|---|---|---|
| **Bacterial Strains** | **Year I** | | **Year II** | |
| | **Plant Height (cm)** | **Root Weight (g)** | **Plant Height (cm)** | **Root Weight (g)** |
| Control | 107.5 c | 1.38 e | 99.74 bc | 1.67 e |
| L-TRP | 110.5 bc | 1.43 de | 107.3 a–c | 1.71 e |
| MN54 | 122.5 a | 1.58 b–d | 117.6 a | 1.92 cd |
| MN54+L-TRP | 127.2 a | 1.62 bc | 118.2 a | 2.02 c |
| MN17 | 110.2 bc | 1.45 c–e | 102.0 bc | 1.72 e |
| MN17+L-TRP | 107.6 c | 1.57 b–d | 103.9 a–c | 1.78 de |
| MN34 | 117.9 ab | 1.91 a | 110.2 ab | 2.36 ab |
| MN34+L-TRP | 119.3 a | 1.96 a | 111.6 ab | 2.50 a |
| PsJN | 94.4 d | 1.72 b | 93.8 c | 2.22 b |
| PsJN+L-TRP | 96.2 d | 1.92 a | 100.7 bc | 2.29 b |

Note: Data represent the means of experiments conducted during two consecutive years, Years I and II. Each experiment was performed with three replications, with each replication comprising ten plants. The means with different letters in a column are significantly different at $p \leq 0.05$ according to Tukey's Test. L-tryptophan; *Enterobacter* sp. MN17; *B. phytofirmans* PsJN; *Pantoea* sp. MN34; and *Bacillus* sp. MN54.

Foliar treatments had a positive effect on number of days taken to anthesis in all types of umbels (Figure 1A–C). MN34 and L-tryptophan induced earliest flowering in primary (54.1 and 52.0 days), secondary (57.0 and 58.6 days), and tertiary umbels (82.6 and 80.0 days), which took 20%, 23%, and 14% less time than the control for all umbel orders, respectively, during both years. The maximum days taken to anthesis in primary (67.0 and 66.0 days), secondary (75.6 and 74.6 days), and tertiary umbels (96.3 and 93.3 days) were observed for the control (untreated) plant (Figure 1).

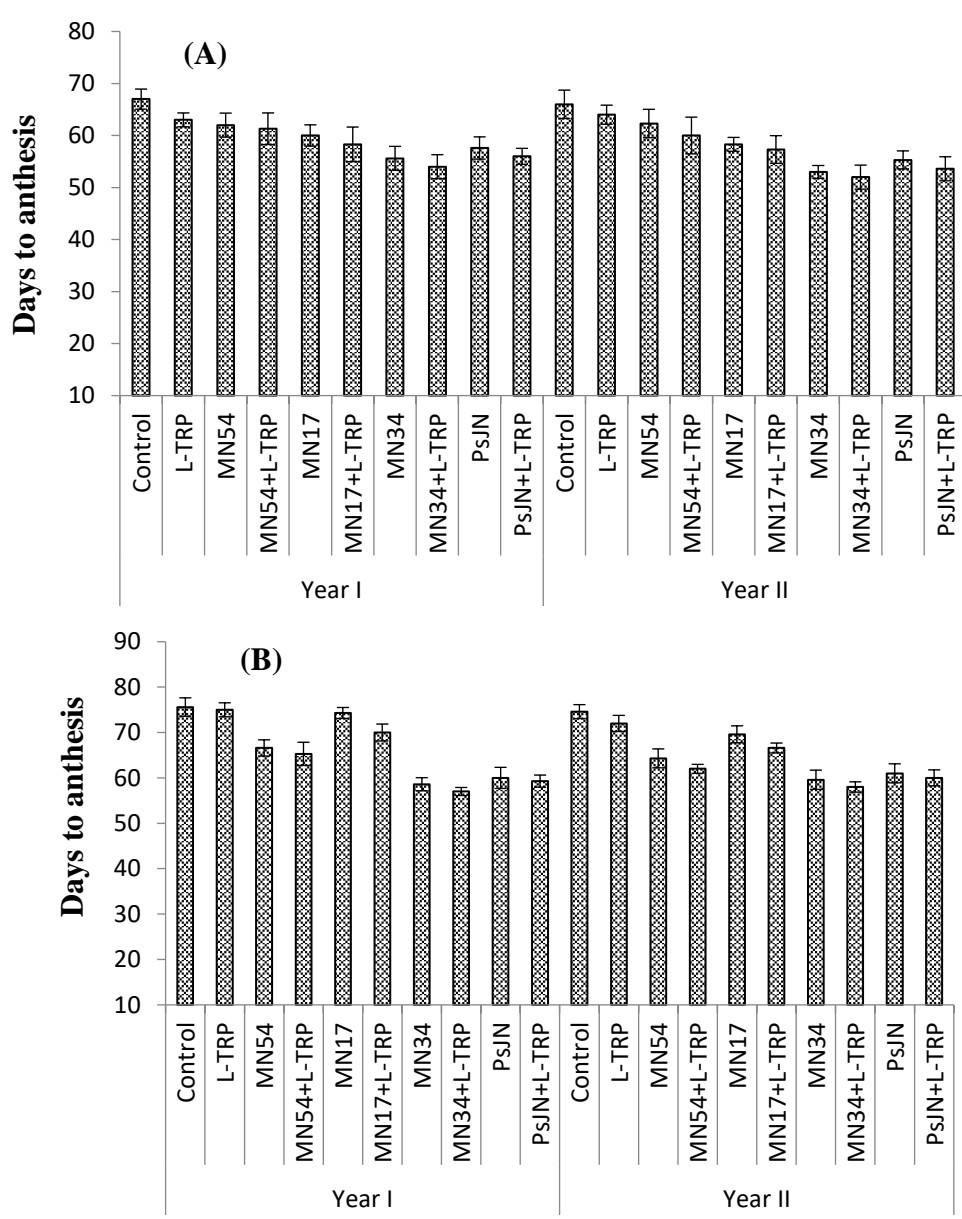

**Figure 1.** *Cont.*

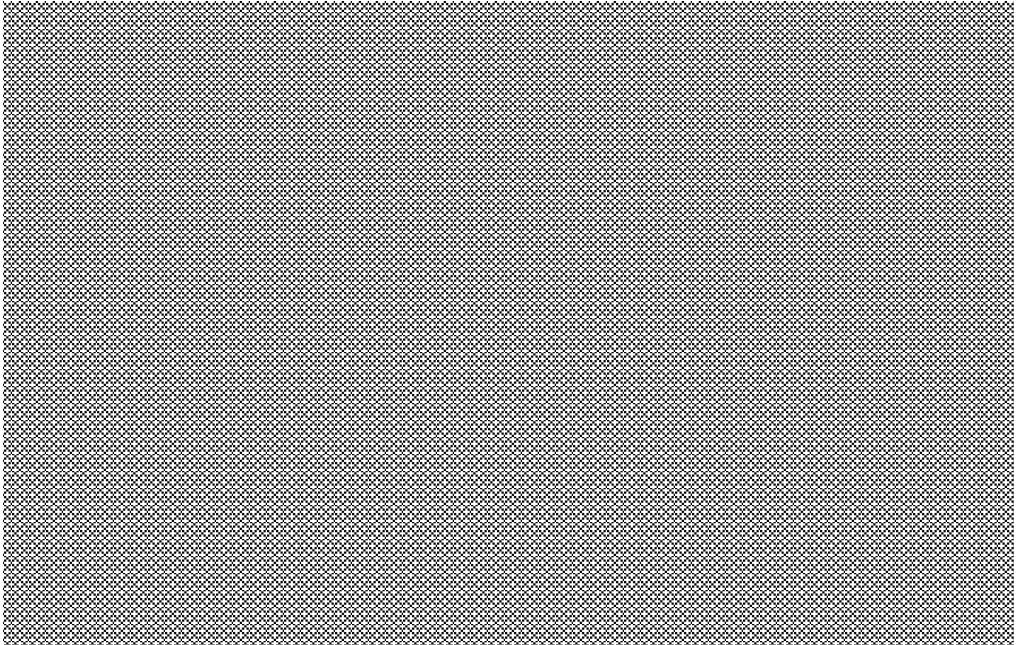

**Figure 1.** The Effects of foliar treatments for days to anthesis in primary umbels (**A**), secondary umbels (**B**), and tertiary umbels (**C**) of carrot cv. T-29 during two consecutive years, Years I and II. Vertical bars indicate means ± SE. *n* = 3. L-tryptophan; *Enterobacter* sp. MN17; *B. phytofirmans* PsJN; *Pantoea* sp. MN34; and *Bacillus* sp. MN54.

The maximum number of secondary umbels per plant (15.0 and 16.0 umbels) was documented with MN17+L-TRP, during Years I and II, respectively, with 40% more secondary umbels per plant recorded in samples treated with MN17+L-TRP, compared with the control (untreated) plants. However, the minimum number of secondary umbels per plant (11 umbels) was found in control during both years (Figure 2A). The least number of tertiary umbels per plant (6.00 and 5.00) was recorded for MN17+L-TRP and MN34+L-TRP, with 45% fewer tertiary umbels per plant recorded compared with the untreated plants during Years I and II. The maximum number of tertiary umbels per plant (10.0 umbels) was recorded for control (untreated) during both years but was statistically similar to L-TRP and MN54 during both years (Figure 2B).

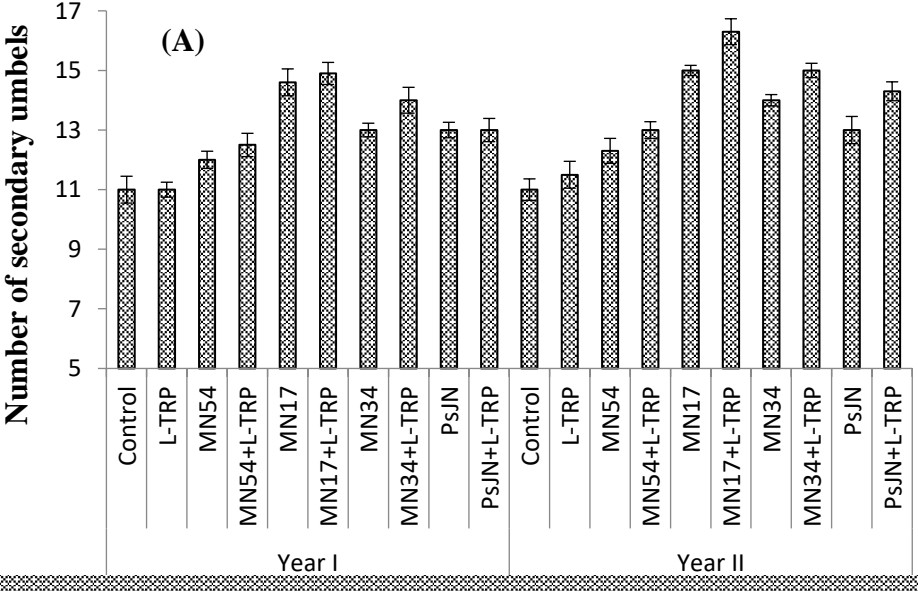

**Figure 2.** *Cont.*

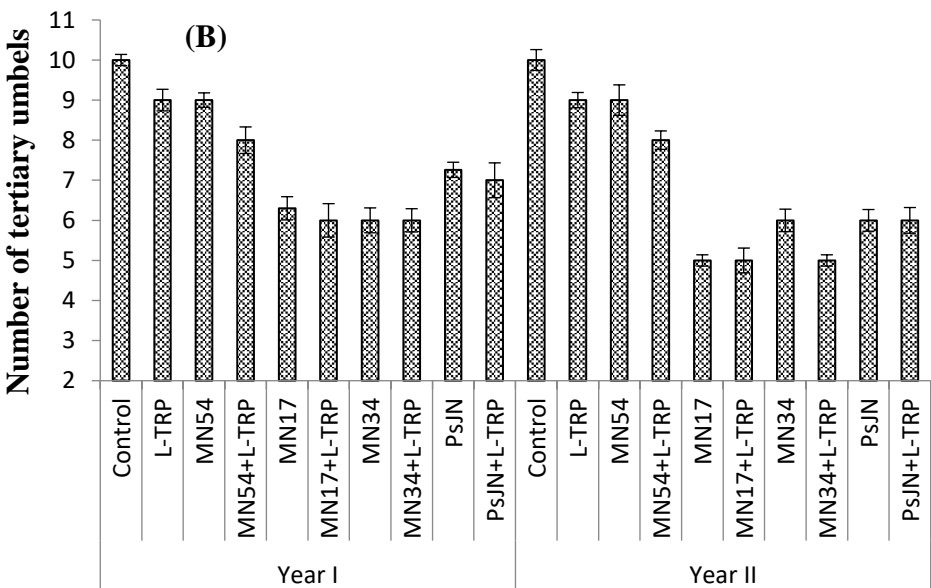

**Figure 2.** The effects of foliar treatments on the number of secondary umbels (**A**) and the number of tertiary umbels (**B**) during two consecutive years, Years I and II. Vertical bars indicate means $\pm$ SE. *n* = 3. L-tryptophan; *Enterobacter* sp. MN17; *B. phytofirmans* PsJN; *Pantoea* sp. MN34; and *Bacillus* sp. MN54.

Seed weights per umbel differed significantly among the foliar treatments and were significantly higher with L-TRP (Table 2). PsJN and L-tryptophan produced higher seed weight from primary (13.7 g and 12.8 g), secondary (56.0 g and 55.6 g), and tertiary umbels (13.1 and 12.7 g), during Years I and II, with 44%, 41%, and 42% higher seed weight recorded in all umbel orders, i.e., primary, secondary, and tertiary umbels, respectively, compared with respective control plants (Table 2). The effects of foliar treatments on seed quality were more variable. A higher value of 1000-seed weight was observed in primary (5.9 and 5.7 g), secondary (5.3 and 5.2 g), and tertiary umbels (2.74 and 2.51 g) for plants sprayed with PsJN and L-TRP, indicating 42%, 34%, and 35% increases in the umbel orders, respectively, compared with the control (untreated) plants during both years (Table 2).

**Table 2.** The effects of bacterial strains on seed weight per umbel and weight of 1000-seed weight from primary, secondary, and tertiary umbels of carrot cv. T-29.

| Treatments | Seed Weight Per Umbel (g) | | | | | |
|---|---|---|---|---|---|---|
| | Year I | | | Year II | | |
| Bacterial Strains | Primary | Secondary | Tertiary | Primary | Secondary | Tertiary |
| Control | 9.5 c | 39.1 e | 9.38 e | 8.9 g | 40.0 f | 8.91 g |
| L-TRP | 9.7 c | 43.5 de | 9.92 de | 9.0 fg | 41.6 ef | 9.21 fg |
| MN54 | 10.0 c | 45.7 cd | 10.4 c–e | 9.31 e–g | 42.9 d–f | 9.92 e–g |
| MN54+L-TRP | 10.8 bc | 46.5 cd | 10.7 cd | 10.2 d–f | 45.8 c–f | 10.3 d–f |
| MN17 | 11.3 a–c | 48.5 b–d | 10.9 cd | 10.5 c–e | 47.6 b–e | 10.5 de |
| MN17+L-TRP | 12.0 a–c | 49.4 b–d | 11.2 bc | 11.2 b–d | 48.6 a–e | 11.0 c–e |
| MN34 | 12.6 ab | 51.6 a–c | 11.6 bc | 11.7 a–c | 50.3 a–d | 11.4 b–d |
| MN34+L-TRP | 13.1 ab | 53.7 ab | 12.1 ab | 12.0 ab | 52.6 a–c | 12.0 a–c |
| PsJN | 13.4 ab | 54.0 ab | 12.9 a | 12.5 a | 54.6 ab | 12.4 ab |
| PsJN+L-TRP | 13.7 a | 56.0 a | 13.1 a | 12.8 a | 55.6 a | 12.7 a |

**Table 2.** *Cont.*

| | 1000-Seed Weight (g) | | | | | |
|---|---|---|---|---|---|---|
| | Year I | | | Year II | | |
| | Primary | Secondary | Tertiary | Primary | Secondary | Tertiary |
| Control | 4.2 f | 3.9 d | 1.97 b | 4.0 e | 3.93 e | 1.91 e |
| L-TRP | 4.5 ef | 3.9 d | 2.24 ab | 4.1 e | 3.97 e | 1.93 e |
| MN54 | 4.7 d–f | 4.1 cd | 2.28 ab | 4.2 de | 4. 01 e | 1.97 d–e |
| MN54+L-TRP | 4.9 c–e | 4.4 b–d | 2.38 ab | 4.60 c–e | 4.1 de | 2.05 c–e |
| MN17 | 5.0 c–e | 4.6 a–d | 2.40 ab | 4.91 b–d | 4.28 c–e | 2.18 b–d |
| MN17+L-TitRP | 5.1 b–e | 4.7 a–c | 2.49 ab | 5.0 a–c | 4.48 cd | 2.29 a–c |
| MN34 | 5.2 b–d | 4.9 ab | 2.54 ab | 5.0 a–c | 4.6 bc | 2.32 ab |
| MN34+L-TRP | 5.4 a–c | 5.1 ab | 2.59 a | 5.3 a–c | 4.92 ab | 2.41 ab |
| PsJN | 5.7 ab | 5.2 a | 2.65 a | 5.4 ab | 5.0 ab | 2.45 a |
| PsJN+L-TRP | 5.9 a | 5.3 a | 2.74 a | 5.71 a | 5.2 a | 2.51 a |

Data represent the means of experiments conducted during two consecutive years, Years I and II. Each experiment was performed in three replications, with each replication comprising ten plants. The means with different letters in a column are significantly different at $p \leq 0.05$ according to Tukey's Test. L-tryptophan; *Enterobacter* sp. MN17; *B. phytofirmans* PsJN; *Pantoea* sp. MN34; and *Bacillus* sp. MN54.

Germination percentage was highest in the primary (98.6% and 99.6%), secondary (95.3% and 97.0%), and tertiary umbels (66.2% and 68.0%) seeds of plants treated with PsJN+L-TRP, and it increased up to 16%, 15% and 44%, respectively, for seeds in all umbel orders, compared with the seeds obtained from untreated plants during both years (Year I and II, respectively) (Table 3). Higher vigor index was observed in the primary (1112.9 and 1150.0), secondary (1085.8 and 1111.0), and tertiary umbels (480.8 and 493.7) of plants treated with PsJN+L-TRP than untreated plants, with 44%, 43%, and 46% higher vigor indices than the seeds obtained from untreated plants for all umbel orders, i.e., primary, secondary and tertiary umbels, respectively, during both years (Table 3).

**Table 3.** Effects of bacterial strains on germination (percentage) and vigor index of seeds from primary, secondary, and tertiary umbels of carrot cv. T-29.

| Treatments | Germination (%) | | | | | |
|---|---|---|---|---|---|---|
| **Bacterial Strains** | Year I | | | Year II | | |
| | Primary | Secondary | Tertiary | Primary | Secondary | Tertiary |
| Control | 84.6 f | 82.6 e | 47.3 f | 85.3 d | 84.5 c | 46.0 g |
| L-TRP | 86.0 f | 84.0 de | 48.6 f | 86.0 cd | 84.5 c | 47.0 fg |
| MN54 | 87.3 ef | 85.6 c–e | 50.6 ef | 91.1 bc | 88.0 bc | 50.0 ef |
| MN54+LTRP | 89.3 d–f | 87.0 b–e | 52.0 d–f | 92.6 b | 89.3 a–c | 52.0 d–f |
| MN17 | 91.3 c–e | 89.3 a–e | 55.3 c–e | 91.1 bc | 91.5 a–c | 56.0 c–e |
| MN17+LTRP | 92.6 b–d | 90.6 a–d | 57.3 b–d | 92.6 b | 93.0 ab | 57.0 cd |
| MN34 | 93.3 b–d | 92.0 a–c | 60.6 a–c | 95.5 ab | 93.3 ab | 60.0 bc |
| MN34+LTRP | 96.3 a–c | 93.3 a–c | 62.0 ab | 95.5 ab | 95.5 ab | 67.5 a |
| PsJN | 97.3 ab | 94.6 ab | 64.0 a | 99.2 a | 96.3 a | 65.0 ab |
| PsJN+L-TRP | 98.6 a | 95.3 a | 66.2 a | 99.6 a | 97.0 a | 68.0 a |
| | Vigor Index | | | | | |
| | Year I | | | Year II | | |
| | Primary | Secondary | Tertiary | Primary | Secondary | Tertiary |
| Control | 764.4 f | 755.5 g | 329.6 f | 778.4 h | 764.0 g | 335.4 h |
| L-TRP | 793.6 f | 771.0 fg | 334.4 f | 813.6 gh | 781.9 fg | 346.9 h |
| MN54 | 816.5 ef | 801.7 f | 355.2 ef | 843.6 fg | 812.1 f | 366.6 g |
| MN54+LTRP | 865.8 de | 845.9 e | 368.9 e | 888.4 ef | 858.3 e | 386.6 f |
| MN17 | 893.6 cd | 868.5 e | 378.9 de | 916.1 de | 904.3 d | 395.6 ef |
| MN17+LTRP | 923.3 c | 911.5 d | 400.0 cd | 947.7 cd | 941.8 cd | 410.0 de |
| MN34 | 1000.6 b | 975.6 c | 411.6 bc | 995.7 c | 985.2 c | 427.3 d |
| MN34+LTRP | 1061.4 ab | 1016.8 b | 464.9 a | 1072.5 b | 1047.0 b | 472.3 b |
| PsJN | 1086.6 a | 1033.7 b | 434.2 b | 1094.9 b | 1061.7 b | 444.8 c |
| PsJN+L-TRP | 1112.9 a | 1085.8 a | 480.8 a | 1150.0 a | 1111.0 a | 493.7 a |

Data represent the means of experiments conducted during two consecutive years, Years I and II. Each experiment was performed in three replications, with each replicate comprising ten plants. The means with different letters in a column are significantly different at $p \leq 0.05$ according to Tukey's Test. L-tryptophan; *Enterobacter* sp. MN17; *B. phytofirmans* PsJN; *Pantoea* sp. MN34; and *Bacillus* sp. MN54.

The lowest conductivity of leachates of harvested seeds, after 24 h of soaking seeds in deionized water, was recorded in the seeds from the primary (427.6 and 372.6 µS/cm), secondary (486.3 and 470.0 µS/cm), and tertiary umbels (1080.3 and 948.7 µS/cm) in response to foliar application of PsJN+L-TRP, i.e., 34.5%, 29.8% and 35.5% lower leachates, respectively, than in the seeds from the corresponding umbels of untreated plants during both years (Table 4). Malondialdehyde contents of the seeds harvested from the primary (1.16 and 1.22 µmols/g Fw), secondary (1.20 and 1.15 µmols/g Fw), and tertiary umbels (3.59 and 3.43 µmols/g Fw) were lowest for plants treated with PsJN+L-TRP, i.e., 36.1%, 40.6%, and 30.1% lower than seeds obtained from all umbel orders of control plants, respectively during both years (Table 4).

**Table 4.** The effects of bacterial strains on electrical conductivity (EC) and malondialdehyde contents (MDA) of seeds from primary, secondary, and tertiary umbel of carrot cv. T-29.

| Treatments | EC (µS/cm) | | | | | |
|---|---|---|---|---|---|---|
| **Bacterial Strains** | Year I | | | Year II | | |
| | Primary | Secondary | Tertiary | Primary | Secondary | Tertiary |
| Control | 633.0 a | 692.6 a | 1594.0 a | 598.0 a | 685.3 a | 1576.6 a |
| L-TRP | 615.0 a | 669.0 ab | 1553.3 a | 571.3 b | 648 ab | 1495.7 a |
| MN54 | 581.0 ab | 664.3 ab | 1475.0 ab | 558.0 b | 644 ab | 1390.3 b |
| MN54+LTRP | 576.0 ab | 615.0 bc | 1386.0 bc | 519.6 c | 598 b | 1365.7 bc |
| MN17 | 531.6 bc | 611.3 bc | 1464.3 ab | 513.7 c | 594 b | 1299.7 c |
| MN17+LTRP | 507.0 c | 574.6 cd | 1376.0 bc | 454.6 d | 535.3 c | 1200 d |
| MN34 | 499.0 cd | 564.6 cd | 1279.0 cd | 445 d | 489 cd | 1188 d |
| MN34+L-TRP | 475.0 c–e | 524.6 de | 1168.0 de | 375.3 f | 480 cd | 1031 ef |
| PsJN | 436.0 de | 514.0 de | 1268. 7 cd | 406 e | 479.3 cd | 1053 e |
| PsJN+L-TRP | 427.6 e | 486.3 e | 1080.3 e | 372.6 f | 470 d | 948.7 f |
| MDA (µmols/g Fw) | | | | | | |
| | Year I | | | Year II | | |
| | Primary | Secondary | Tertiary | Primary | Secondary | Tertiary |
| Control | 1.74 a | 1.89 a | 5.01 a | 2.01 a | 2.08 a | 5.02 a |
| L-TRP | 1.65 ab | 1.78 ab | 4.82 ab | 1.74 b | 1.91 a | 4.81 ab |
| MN54 | 1.57 ab | 1.64 bc | 4.72 a–c | 1.62 bc | 1.71 b | 4.62 bc |
| MN54+LTRP | 1.54 ab | 1.61 cd | 4.46 b–d | 1.63 bc | 1.63 b | 4.54 b–d |
| MN17 | 1.45 bc | 1.52 c–e | 4.23 c–e | 1.52 b–d | 1.58 bc | 4.39 cd |
| MN17+LTRP | 1.38 b–d | 1.46 d–f | 4.02 d–f | 1.43 c–e | 1.53 bc | 4.26 d |
| MN34 | 1.25 cd | 1.36 e–g | 3.81 ef | 1.31 de | 1.41 cd | 3.90 e |
| MN34+L-TRP | 1.21 cd | 1.31 fg | 3.80 ef | 1.26 e | 1.29 de | 3.77 e |
| PsJN | 1.21 cd | 1.23 g | 3.69 f | 1.23 e | 1.21 e | 3.69 ef |
| PsJN+L-TRP | 1.16 d | 1.20 g | 3.59 f | 1.22 e | 1.15 e | 3.43 f |

Data represent the means of experiments conducted during two consecutive years, Years I and II. Each experiment was conducted in three replications, with each replicate comprising ten plants. The means with different letters in a column are significantly different at $p \leq 0.05$ according to Tukey's Test. L-tryptophan; *Enterobacter* sp. MN17; *B. phytofirmans* PsJN; *Pantoea* sp. MN34; and *Bacillus* sp. MN54.

The activity of antioxidant enzymes in the harvested seeds of all types of umbels was also positively affected by the foliar application of plant-growth-promoting bacterial strains. The superoxide dismutase activity of seedlings raised from the primary (431.8 and 453.8 U kg$^{-1}$ protein), secondary (420.4 and 435.1 U kg$^{-1}$ protein), and tertiary umbels (254.3 and 266.8 U kg$^{-1}$ protein) was highest in plants treated with PsJN+L-TRP, increasing up to 46.5%, 44.1%, and 41.5%, respectively, compared with untreated (control) plants (Table 5). A higher value of peroxidase activity of seedlings raised from the primary (1193.1 and 1256.6 U kg$^{-1}$ protein), secondary (1142 and 1085.7 U kg$^{-1}$ protein), and tertiary umbels (600.1 and 615.9 U kg$^{-1}$ protein) was observed for plants treated with PsJN+L-TRP, while it was found to be minimum for control, i.e., the plants treated with PsJN+L-TRP

exhibited 43.5%, 41.5% and 44% higher peroxidase activity than the untreated (control) plants in all umbel orders, respectively (Table 5).

**Table 5.** The effects of bacterial strains on superoxide dismutase (SOD) and peroxidase activity (POD) of seeds from primary, secondary, and tertiary umbel of carrot cv. T-29.

| Treatments | SOD (U kg$^{-1}$ Protein) | | | | | |
|---|---|---|---|---|---|---|
| | Year I | | | Year II | | |
| **Bacterial Strains** | Primary | Secondary | Tertiary | Primary | Secondary | Tertiary |
| Control | 297.4 g | 292.6 g | 180.6 f | 304.7 f | 300.2 f | 185.9 h |
| L-TRP | 300.4 g | 298.5 g | 187.9 ef | 311.0 f | 308.6 f | 191.7 gh |
| MN54 | 311.2 fg | 305.9 fg | 198.9 de | 326.9 ef | 317.2 ef | 200.5 fg |
| MN54+L-TRP | 324.7 ef | 317.1 ef | 205.1 d | 352.6 de | 338.9 de | 209.8 ef |
| MN17 | 339.3 e | 332.8 de | 211.5 cd | 371.3 d | 356.1 d | 216.5 e |
| MN17+L-TRP | 366.1 d | 349.8 cd | 220.5 c | 400.1 c | 391.7 c | 233.2 d |
| MN34 | 385.4 cd | 362.2 c | 233.9 b | 413.0 c | 400.6 c | 244.8 d |
| MN34+L-TRP | 403.4 bc | 398.1 b | 249.4 a | 446.5 ab | 426.0 ab | 258.7 ab |
| PsJN | 417.2 ab | 408.5 ab | 242.9 ab | 423.8 bc | 411.5 bc | 254.1 bc |
| PsJN+L-TRP | 431.8 a | 420.4 a | 254.3 a | 453.8 a | 435.1 a | 266.8 a |
| | POD (U kg$^{-1}$ Protein) | | | | | |
| | Year I | | | Year II | | |
| | Primary | Secondary | Tertiary | Primary | Secondary | Tertiary |
| Control | 841.7 g | 812.3 f | 417.1 g | 862.3 g | 759.5 g | 424.1 g |
| L-TRP | 855.0 g | 826.4 f | 427.9 fg | 873.8 g | 765.6 g | 437.3 fg |
| MN54 | 874.7 fg | 848.7 ef | 442.0 fg | 899.6 g | 800.3 fg | 458.3 e–g |
| MN54+L-TRP | 905.8 ef | 861.0 ef | 453.6 ef | 916.7 fg | 832.1 ef | 473.6 d–f |
| MN17 | 931.8 e | 901.4 e | 477.4 de | 980.7 ef | 880.2 de | 484.9 c–e |
| MN17+L-TRP | 986.6 d | 967.4 c | 494.7 cd | 1040.7 de | 925.0 cd | 503.7 cd |
| MN34 | 1053.6 c | 993.8 cd | 521.2 c | 1101.9 cd | 990.7 bc | 522.7 c |
| MN34+L-TRP | 1099.1 bc | 1039.7 bc | 563.0 b | 1140.9 bc | 1050.7 ab | 572.9 b |
| PsJN | 1145.7 ab | 1087.7 ab | 583.7 ab | 1205.9 ab | 1059.8 a | 593.3 ab |
| PsJN+L-TRP | 1193.1 a | 1142.0 a | 600.1 a | 1256.6 a | 1085.7 a | 615.9 a |

Data represent the means of experiments conducted during two consecutive years, Years I and II. Each experiment was conducted in three replications, with each replicate comprising ten plants. The means with different letters in a column are significantly different at $p \leq 0.05$ according to Tukey's Test. L-tryptophan; *Enterobacter* sp. MN17; *B. phytofirmans* PsJN; *Pantoea* sp. MN34; and *Bacillus* sp. MN54.

The highest catalase activity of seedlings raised from the primary (601.3 and 616.3 U kg$^{-1}$ protein), secondary (588.1 and 600.8 U kg$^{-1}$ protein), and tertiary umbels (377.7 and 365.1 U kg$^{-1}$ protein) was recorded for plants treated with PsJN+L-TRP, with 49%, 47.9%, and 46% increase in catalase activity compared with untreated (control) plants (Table 6). The total antioxidant content in the seedlings of the primary (99.5% and 99% inhibition) secondary (96.1% and 93.9% inhibition), and tertiary umbels (72.9% and 63.6% inhibition) were recorded highest for plants treated with MN34+L-TRP, increasing up to 32.4%, 40.7%, and 47.7%, respectively, compared with untreated (control) plants in all umbel orders. The least value for total antioxidant content was recorded for the control (Table 6).

**Table 6.** The effects of bacterial strains on catalase activity (CAT) and total antioxidant content (TAC) of seeds from primary, secondary, and tertiary umbels of carrot cv. T-29.

| Treatments | CAT (U kg$^{-1}$ Protein) | | | | | |
|---|---|---|---|---|---|---|
| **Bacterial Strains** | Year I | | | Year II | | |
| | Primary | Secondary | Tertiary | Primary | Secondary | Tertiary |
| Control | 400.8 f | 398.1 f | 260.7 d | 416.5 f | 405.2 g | 247.2 f |
| L-TRP | 410.2 f | 405.2 f | 267.1 d | 428.7 f | 413.6 fg | 253.7 ef |
| MN54 | 430.1 ef | 415.0 ef | 286.0 cd | 438.4 ef | 433.5 fg | 261.7 ef |
| MN54+L-TRP | 470.1 cd | 428.3 ef | 292.8 d | 465.3 de | 453.0 ef | 272.4 de |
| MN17 | 451.3 de | 450.2 de | 300.5 c | 495.8 cd | 485.1 de | 285. 7 cd |
| MN17+L-TRP | 503.2 bc | 488.1 cd | 312.1 c | 522.3 c | 504.8 cd | 300.6 c |
| MN34 | 535.1 b | 526.7 bc | 343.8 b | 563.3 b | 542.6 bc | 332.1 b |
| MN34+L-TRP | 586.9 a | 548.3 ab | 365.8 ab | 601.1 a | 593.3 a | 359.4 a |
| PsJN | 595.4 a | 576.6 a | 346.5 b | 593.8 ab | 565.8 ab | 348.3 ab |
| PsJN+L-TRP | 601.3 a | 588.1 a | 377.7 a | 616.3 a | 600.8 a | 365.1 a |
| | **TAC (% Inhibition)** | | | | | |
| | Year I | | | Year II | | |
| | Primary | Secondary | Tertiary | Primary | Secondary | Tertiary |
| Control | 76.8 e | 65.6 f | 49.5 d | 73.2 e | 69.6 e | 42.8 f |
| L-TRP | 80.3 de | 67.3 ef | 50.8 cd | 75.0 e | 71.7 e | 44.1 ef |
| MN54 | 81.2 de | 71.0 e | 53.9 cd | 80.0 de | 73.7 de | 46.4 ef |
| MN54+L-TRP | 82.6 cd | 76.2 d | 55.9 bc | 82.6 d | 74.0 de | 47.1 ef |
| MN17 | 86.7 bc | 77.5 d | 59.9 b | 86.7 cd | 78.5 c–e | 49.3 de |
| MN17+L-TRP | 87.9 b | 83.8 c | 60.3 b | 90.1 bc | 81.9 b–d | 53.2 cd |
| MN34 | 96.6 a | 94.9 a | 70.8 a | 97.8 a | 89.6 ab | 58.4 a–c |
| MN34+L-TRP | 99.5 a | 96.1 a | 72.9 a | 99.0 a | 93.9 a | 63.6 a |
| PsJN | 95.2 a | 85.7 bc | 67.5 a | 95.2 ab | 84.9 a–c | 56.3 bc |
| PsJN+L-TRP | 99.0 a | 89.6 b | 68.5 a | 96.6 ab | 87.9 ab | 61.6 ab |

Data represent the means of experiments conducted during two consecutive years, Years I and II. Each experiment was performed in three replications, with each replication comprising ten plants. The means with different letters in a column are significantly different at $p \leq 0.05$ according to Tukey's Test. L-tryptophan; *Enterobacter* sp. MN17; *B. phytofirmans* PsJN; *Pantoea* sp. MN34; and *Bacillus* sp. MN54.

The total phenolic content in the seedlings of primary (295.5 and 300.7 mg GAE 100 g$^{-1}$), secondary (264.5 and 274.2 mg GAE 100 g$^{-1}$), and tertiary (189.0 and 202.9 mg GAE 100 g$^{-1}$) umbels of plants treated with PsJN+L-TRP were increased up to 42.3%, 35.3%, and 45.5%, respectively, compared with the control (Table 7).

**Table 7.** The effects of bacterial strains on total phenolic content (TPC) of seeds from primary, secondary, and tertiary umbels of carrot cv. T-29.

| Treatments | TPC (mg GAE 100 g$^{-1}$) | | | | | |
|---|---|---|---|---|---|---|
| **Bacterial Strains** | Year I | | | Year II | | |
| | Primary | Secondary | Tertiary | Primary | Secondary | Tertiary |
| Control | 205.6 g | 198.7 e | 130.6 f | 213.7 f | 199.3 f | 138.5 g |
| L-TRP | 212.8 fg | 204.5 e | 132.0 f | 220.9 ef | 212.3 ef | 141.8 fg |
| MN54 | 247.9 c–e | 209.3 de | 138.3 ef | 227.7 ef | 220.8 e | 153.2 ef |
| MN54+LTRP | 250.7 cd | 213.3 de | 143.9 de | 245.1 de | 225.9 de | 155.8 de |
| MN17 | 227.7 e–g | 219.3 c–e | 145.9 de | 260.4 cd | 229.0 de | 167.3 cd |
| MN17+LTRP | 229.6 d–f | 233.6 b–d | 151.0 cd | 264.5 b–d | 241.8 cd | 172.7 c |
| MN34 | 266.3 bc | 244.2 a–c | 161.4 bc | 280.7 a–c | 248.4 bc | 188.6 b |
| MN34+LTRP | 270.5 bc | 249.3 ab | 188.8 a | 290.0 ab | 263.8 ab | 197.1 ab |
| PsJN | 285.7 ab | 262.0 a | 172.7 b | 287.4 ab | 267.5 a | 189.4 ab |
| PsJN+L-TRP | 295.5 a | 264.5 a | 189.0 a | 300.7 a | 274.2 a | 202.9 a |

Data represent the means of experiments conducted during two consecutive years, Years I and II. Each experiment was performed in three replications, with each replication comprising ten plants. The means with different letters in a column are significantly different at $p \leq 0.05$ according to Tukey's Test. L-tryptophan; *Enterobacter* sp. MN17; *B. phytofirmans* PsJN; *Pantoea* sp. MN34; and *Bacillus* sp. MN54.

The bacterial strain colonized the rhizosphere and interior of carrot roots and shoots (Table 8). When augmented with L-TRP, PsJN resulted in enhanced colonization as compared to sole application in the rhizosphere and plant tissues. During the first year of inoculation treatment, the colonization values were $5.70 \times 10^5$ CFU g$^{-1}$ in rhizosphere soil, $2.5 \times 10^5$ CFU g$^{-1}$ in the root interior, and $5.1 \times 10^5$ CFU g$^{-1}$ in the shoot tissue. Similarly, in the second year, the L-TRP-treated rhizosphere soil CFU g$^{-1}$ value was $6.8 \times 10^5$, whereas in the root interior, it was $3.71 \times 10^5$, and in the shoot interior, it was $7.8 \times 10^5$. However, more CFU of the inoculant strain g$^{-1}$ dry weight were found from the rhizosphere and the interior of roots and shoots treated with L-TRP.

**Table 8.** The effects of bacterial strains on the persistence of bacterial isolates (CFU $\times 10^5$ g$^{-1}$ dry weight) in the rhizosphere, root interior, and shoot interior of carrot cv. T-29.

| Treatments | CFU $\times 10^5$ g$^{-1}$ Dry Weight | | | | | |
|---|---|---|---|---|---|---|
| | Year I | | | Year II | | |
| Bacterial Strains | Rhizosphere | Root | Shoot | Rhizosphere | Root | Shoot |
| L-TRP | 4.11 e | 1.92 e | 3.96 e | 5.4 ef | 3.23 ab | 4.8 d |
| MN54 | 4.21 de | 1.96 d–e | 4. 02 e | 5.6 d–f | 3.27 ab | 5.0 cd |
| MN54+L-TRP | 4.62 c–e | 2.06 c–e | 4.13 de | 5.8 c–e | 3.37 ab | 5.3 b–d |
| MN17 | 4.92 b–d | 2.17 b–d | 4.27 c–e | 6.1 c–e | 3.39 ab | 5.5 a–d |
| MN17+L-TRP | 5.0 a–c | 2.28 a–c | 4.47 cd | 6.2 b–e | 3.48 ab | 5.6 a–c |
| MN34 | 5.01 a–c | 2.31 ab | 4.5 bc | 6.1 b–d | 3.55 ab | 5.8 ab |
| MN34+L-TRP | 5.2 a–c | 2.42 ab | 4.91 ab | 6.3 a–c | 3.58 a | 6.2 ab |
| PsJN | 5.3 ab | 2.44 a | 5.02 ab | 6.6 ab | 3.64 a | 6.9 a |
| PsJN+L-TRP | 5.70 a | 2.50 a | 5.1 a | 6.8 a | 3.73 a | 7.2 a |

Data represent the means of experiments conducted during two consecutive years, Years I and II. Each experiment was performed in three replications, with each replication comprising ten plants. The means with different letters in a column are significantly different at $p \leq 0.05$ according to Tukey's Test. L-tryptophan; *Enterobacter* sp. MN17; *B. phytofirmans* PsJN; *Pantoea* sp. MN34; and *Bacillus* sp. MN54.

Data in Table 9 show that PsJN produced auxin (IAA equivalents) without L-TRP application; however, IAA equivalents significantly increased when the medium was applied with L-TRP. In the second year, PsJN produced maximum IAA equivalents of 12.7 µg mL$^{-1}$ when L-TRP was added.

**Table 9.** The effects of bacterial strains on auxin production in the rhizosphere of carrot cv. T-29.

| | Auxin Production (IAA Equivalent µmL$^{-1}$) | | | | |
|---|---|---|---|---|---|
| Treatments | Without L-TRP | | Treatments | With L-TRP | |
| Bacterial Strains | Year 1 | Year 2 | Bacterial Strains | Year 1 | Year 2 |
| Control | 0.26 f | 0.38 g | L-TRP | 3.1 ef | 4.43 fg |
| MN54 | 0.44 d–f | 0.62 e–g | MN54+L-TRP | 4.9 c–e | 10.3 d–f |
| MN17 | 0.53 c–e | 0.76 de | MN17+L-TRP | 5.1 b–e | 11.0 c–e |
| MN34 | 0.62 b–d | 0.82 b–d | MN34+L-TRP | 5.4 a–c | 12.0 a–c |
| PsJN | 0.71 ab | 0.9 ab | PsJN+L-TRP | 5.9 a | 12.7 a |

Data represent the means of experiments conducted during two consecutive years, Years I and II. Each experiment was conducted in three replications, with each replication comprising ten plants. The means with different letters in a column are significantly different at $p \leq 0.05$ according to Tukey's Test. L-tryptophan; *Enterobacter* sp. MN17; *B. phytofirmans* PsJN; *Pantoea* sp. MN34; and *Bacillus* sp. MN54.

## 4. Discussion

Plant-growth-promoting bacteria colonize the rhizosphere and interior tissues of host plants and promote the growth and yield of crops through different mechanisms [53]. In the current study, we found that auxin production by strain *B. phytofirmans* PsJN enhances plant growth via aggressive colonization.

Understanding the impact of agricultural technologies on soil microbiome and enzymatic activity is essential for the quantitative and qualitative assessment of agricultural production [54]. In the present study, it was found that PsJN inoculation improved carrot growth, physiology, root biomass, biochemical activities, and yield traits as compared to the uninoculated control (Tables 1–3 and 5–7). Many bacterial mechanisms have been suggested, and hormone production is known as the most important mechanism in regulating plant growth and development [55]. Among phytohormones, indole-3-acetic acid (IAA) produced by bacteria is a major naturally occurring phytohormone that is also exogenously applied to improve plant growth [40,56–58]. Axillary buds are inhibited by IAA, and shoot apical dominance is induced [59,60]. Microorganisms with tryptophan supplementation in culture media increase IAA production; however, in the absence of L-tryptophan (L-TRP), a small number of auxins are produced [21,40,61,62]. Therefore, in the present study, the impact of PGPB with or without L-TRP on auxin-induced apical dominance and alteration of umbel order was studied in carrot.

The foliar application of plant-growth-promoting bacteria (PGPB) positively influenced various growth traits, yield, and seed-quality attributes. The foliar application of MN54+L-TRP increased the height of carrot seed plants, which is in line with the findings of Hassan and Bano [63], who reported that the inoculation of *Pseudomonas moraviensis* and *Bacillus cereus* with tryptophan addition increased the plant height of wheat grown in pots and fields. An increase in plant height with PGPB and L-tryptophan may be responsible for enhanced IAA availability, which induces cell division and cell elongation [64,65].

Auxin-producing bacterial strains stimulated root growth in this study, as evidenced by the increased weight of adventitious roots in response to foliar applications of MN34 and PsJN with L-tryptophan. Naveed et al. [40] also reported that maize seed inoculation with PsJN supplemented with L-TRP notably augmented root biomass compared with the uninoculated control. Moreover, a positive link between IAA production and root elongation in response to PGPR in lentils has also been reported, which reflects our results [66]. This increase in root growth might also have enhanced nutrient uptake, one of the roles attributed to PGPB [64,65]. This supposition is based on a 30% and 27% increase in seed yield from secondary umbels, while a 26% and 22% increase was observed in 1000-seed weight from secondary umbels in response to the foliar application of *Burkholderia phytofirmans* PsJN and *Pantoea* sp. MN34 along with L-TRP. PsJN has been reported to increase endogenous IAA in maize along with higher nitrogen and phosphorus contents and photosynthetic rate, ultimately resulting in higher biomass [40]. This increase in seed yield with concomitant improvement in yield can also be explained by the findings of Sun et al. [67], who envisaged that the overexpression of the auxin response factor 19 gene caused improvement in the seed size and yield of *Arabidopsis thaliana* and *Jatropha curcas*. Very recently, Iqbal et al. [68] revealed microbial inoculation-mediated enhancement in the growth and yield of canola through auxin production and nutrient uptake. They further reported that the coinoculation of *Bacillus* sp. MN54 and *Piriformospora indica* significantly enhanced root growth, most notably through high rhizosphere auxin production, which consequently enhanced biomass and nutrient uptake of canola. Enhanced yield and yield-related components in canola and peanut have also been attributed to increased water and nutrient absorption, followed by improved photosynthesis, leading to more assimilation and thus an increase in seed yield [69,70].

Seeds in primary and secondary umbels reach maturity under mild temperatures, whereas during flowering, seed setting, and seed maturation in tertiary umbels, temperatures remain elevated (ranging from 33 to 42.5 °C). Consequently, this has negative impacts on seed yield and quality, including germination and the weight of a thousand seeds. Various researchers have documented this irregular seed maturation across different umbels [13,16,17]. To obtain high-quality seeds, the suppression of tertiary umbels seems to be an option but seems impossible if performed manually. The use of auxins has been suggested in a previous study that resulted in better suppression of tertiary umbels in carrot [18]. Therefore, in this study, the impact of PGPB application on the suppression of

tertiary umbels per plant was evaluated. The foliar application of plant-growth-promoting bacterial strains, i.e., *Enterobacter* sp. MN17 and *Pantoea* sp., with and without L-TRP, significantly reduced the number of tertiary umbels compared with control in carrot seed crop; reductions of 41–47% and 48–50% were observed in the tertiary umbel count using MN17 with and without L-TRP, compared with 37–39% and 42–43% reductions using MN34 with and without L-TRP, respectively. The reduction in tertiary umbels by these PGPB may be because of the shoot apical dominance induced by indole-3-acetic acid, and apical auxin might have inhibited tertiary umbels (which appear as side branches on the secondary branches of umbels, possibly due to an increase in the endogenous level of auxin) [59,71]. This apical dominance phenomenon in carrot seed crops had also been observed when exogenously applied auxins caused significant suppression of tertiary umbels and an increase in the number of secondary umbels per plant [18]. Furthermore, the foliar application of MN34+L-TRP and PsJN+L-TRP induced the earliest anthesis in primary, secondary, and tertiary umbels. Reduced time for the onset of flowering was observed in response to the inoculation of strain MN17 in maize cultivars [21].

The results revealed that the exogenous application of PsJN and MN34 along with L-TRP significantly improved the germination and seedling growth of carrot seeds harvested from primary, secondary, and tertiary umbels. Previously, treatments with plant-growth-promoting bacteria were found to increase germination percentage, seedling vigor, emergence, plant stand, and root and shoot growth in various agronomic and horticultural crops [21,72,73]. IAA-producing rhizobacteria triggered faster germination and induced a higher vigor index, which is an indicator of the state of the health of seedlings and ultimately the productivity of crops [74]. This high germination and vigor index may be due to the better production and metabolism of auxin by plant-growth-promoting bacteria (PGPB) in the presence of L-TRP, which stimulates cellular division [75].

The electrical conductivity test measures the amount of electrolyte leakage from seeds during imbibition and thus reflects damage to seed membranes. An increase in conductivity has been found to be correlated with a decrease in seed quality. The leakage of electrolytes from small and physiologically immature seeds is due to the reorganization of membrane components and conformational changes occurring in cell membranes [76]. The level of lipid peroxidation, expressed as malondialdehyde content, has also been used as an indicator of damage to cell membranes caused by free radicals [77]. Hence, lower MDA content indicates enhanced antioxidant potential, providing improved tolerance to oxidative stress [78]. The lowest electrical conductivity and MDA content of seeds from primary, secondary, and tertiary umbels were found in response to the foliar application of selected bacteria strains (PsJN and MN34) with L-TRP, suggesting that the seeds from these treatments were physiologically better than untreated seeds. Moreover, the findings of this study also revealed that the activity of the investigated antioxidant enzymes, the total antioxidant content, and the total phenolic content in seedlings increased in response to all plant-growth-promoting bacteria, particularly with PsJN and L-TRP. Enzymatic activity is the result of accumulated enzyme action as well as proliferating microorganisms [54]. Previously, researchers [79,80] observed enhanced activities of different ROS-scavenging enzymes in PGPB-inoculated tomato and potato plants possibly because of the detoxification of reactive oxygen species by PGPB.

A successful microbial inoculant colonized the external and/or internal part of plant tissues and established a compatible interaction with the host, in addition to persisting in the soil against microorganisms living in the environment through its rhizocompetence traits [81]. Plant-growth-promoting bacterial inoculants colonize the plant initially, but their persistence over time is not guaranteed. Bacterial invasions in the plant body are difficult, and to improve their efficacy, plant-growth-promoting bacteria were applied three times to ensure their proper invasion in the plant body. Measuring the persistence of microbial inoculants in soil poses technical difficulties, as the inoculant needs to be identified. The tracking and monitoring of the persistence of plant-growth-promoting bacteria released in the environment need to be studied to understand their behavior in

soil and determine which factors influence their survival under various conditions. In the present study, the total bacterial count was measured from the rhizosphere using a culture-dependent approach (Table 9). The growth promotion and quality carrot seed production in the present study were most probably because of the survival of the inoculant strain and its L-TRP-mediated auxin product in the plant rhizosphere. Overall, the MN-34 and PsJN strains along with L-TRP improved growth and quality of carrot seed production.

## 5. Conclusions

In this study, we concluded that PGPB application improved carrot umbels and their order, which ultimately increased seed quality and yield-related attributes of carrot. Interestingly, the combined use of PGPB and L-TRP efficiently colonized the rhizosphere and plant tissues and enhanced the production of auxin by selected strains, resulting in better carrot growth and development. This study suggests that auxin production by bacterial strains (particularly MN34+L-tryptophan) suppresses the tertiary umbels and promoted carrot growth. Hence, the combined use of auxin-producing PGPB and a precursor (L-TRP) could be a novel approach for good-quality carrot seed production on a commercial scale.

**Supplementary Materials:** The following supporting information can be downloaded at: https://www.mdpi.com/article/10.3390/horticulturae9090954/s1, Figure S1: Pictorial presentation of the effect of different plant-growth-promoting bacteria (PGPB) with L-tryptophan on carrot flowering, seed setting, and mature umbels.

**Author Contributions:** Conceptualization, K.Z., M.N. and A.I.K.; methodology, M.N.; software, K.S.K.; validation, K.Z., M.A.G. and I.A.; formal analysis, K.S.K., and R.A.; investigation, A.S. and A.I.K.; resources, M.N. and M.H.S.; data curation, A.N.; writing—original draft preparation, A.N.; writing—review and editing, M.N., M.H.S. and K.Z.; visualization, I.A. and M.A.G.; supervision, K.Z.; project administration, M.N.; funding acquisition, M.H.S. All authors have read and agreed to the published version of the manuscript.

**Funding:** This study was supported by the Researchers Supporting Project number (RSP2023R347), King Saud University, Riyadh, Saudi Arabia.

**Data Availability Statement:** Not applicable.

**Acknowledgments:** The authors highly acknowledge the Endowment Fund Secretariat (EFS), University of Agriculture, Faisalabad, for funding this study, with Project No. 574, "Dissemination of Seed Production Technology of Important Crops (Component I: Vegetable Seed Production)". The authors would like to extend their sincere appreciation to the Researchers Supporting Project number (RSP2023R347), King Saud University, Riyadh, Saudi Arabia.

**Conflicts of Interest:** The authors declare no conflict of interest.

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
