# Peer review of "L-Tryptophan-Dependent Auxin-Producing Plant-Growth-Promoting Bacteria Improve Seed Yield and Quality of Carrot by Altering the Umbel Order"

_horticulturae, doi:10.3390/horticulturae9090954_

Round 1

Reviewer 1 Report

This study reported that L-Tryptophan-dependent auxin producing plant growth promoting bacteria improve seed yield and quality of carrot by altering the umbel order. The results suggested that carrot seed producers may use selected PGPB strains to suppress tertiary umbels and obtain a higher yield of good quality carrot seed. The results of this study have high reference value for the production of carrot seeds.  My comments for this paper as follows:

1. title

The title is very specific and appropriate.

2. Abstract

The abstract provides a good summary of the entire text. Some suggestions are as follows:

1) line 24: Please provide the full name of "PGPB".

2) Line 26: Please provide the full name of 'L-TRP'

3. Keyword

I think 10 keywords are too many, and 6-8 keywords may be better.

4. Introduction

The introduction provides a good overview of the background of this study. Some suggestions are as follows:

Line 72: should there  be a "," between Enterobacter and Alcaligenes?

Line 88: I suggest providing the full name of 'L-TRP'

Line 92: "B. Subtilis" should be italicized

Line 94: Who have reported?

5. Materials and Methods

The research plan is reasonable and the description is detailed.

6. Results

The data is sufficient and the expression is reasonable. All the tabels and figs are good.

7. Discussion and Conclusions

The discussion was thorough and the conclusion was reasonable.

This study reported that L-Tryptophan-dependent auxin producing plant growth promoting bacteria improve seed yield and quality of carrot by altering the umbel order. The results suggested that carrot seed producers may use selected PGPB strains to suppress tertiary umbels and obtain a higher yield of good quality carrot seed. The results of this study have high reference value for the production of carrot seeds.  My comments for this paper as follows:

1. title

The title is very specific and appropriate.

2. Abstract

The abstract provides a good summary of the entire text. Some suggestions are as follows:

1) line 24: Please provide the full name of "PGPB".

2) Line 26: Please provide the full name of 'L-TRP'

3. Keyword

I think 10 keywords are too many, and 6-8 keywords may be better.

4. Introduction

The introduction provides a good overview of the background of this study. Some suggestions are as follows:

Line 72: should there  be a "," between Enterobacter and Alcaligenes?

Line 88: I suggest providing the full name of 'L-TRP'

Line 92: "B. Subtilis" should be italicized

Line 94: Who have reported?

5. Materials and Methods

The research plan is reasonable and the description is detailed.

6. Results

The data is sufficient and the expression is reasonable. All the tabels and figs are good.

7. Discussion and Conclusions

The discussion was thorough and the conclusion was reasonable.

Author Response

Response to Reviewer comments 1

We have read your valuable comments and revised the manuscript according to our best. Please find the attached revised version, which we would like to submit for your kind consideration.

We would like to express our great appreciation to you for comments on our paper. Looking forward to hearing from you.

The questions raised by reviewer and answers are as follows:

Question 1: Title

The title is very specific and appropriate.

Response: Thank you for your nice comment.

Question 2: Abstract

The abstract provides a good summary of the entire text. Some suggestions are as follows:

1) Line 24: Please provide the full name of "PGPB".

2) Line 26: Please provide the full name of 'L-TRP'

Response:

  • Full name of “Plant Growth Promoting Bacteria” (PGPB) have been added in line “24”
  • Full name of “L-Tryptophan” (L-TRP) have been added in line “26”

Question 3: Keyword

I think 10 keywords are too many, and 6-8 keywords may be better.

Response: Thank you for your nice suggestion.

We have added only 8 key words as per your suggestion.

Question 4: Introduction

The introduction provides a good overview of the background of this study. Some suggestions are as follows:

Line 72: should there be a "," between Enterobacter and Alcaligenes?

Line 88: I suggest providing the full name of 'L-TRP'

Line 92: "B. Subtilis" should be italicized

Line 94: Who have reported?

Response:

Thank you for your kind words.

  • Yes, we think it would be more appropriate to add "," between Enterobacter and Alcaligenes and have been incorporated in line 72.
  • The full name of “L-Tryptophan” (L-TRP) has been provided in Line “88.

  • "B. Subtilis" has been italicized in Line 92.

  • Noor et al., 2020” have reported their work.

Question 5: Materials and Methods

The research plan is reasonable and the description is detailed.

Response: Thank you for nice comments.

Question 6: Results

The data is sufficient and the expression is reasonable. All the tables and figs are good.

Response: Thank you so much for taking the time to leave us feedback.

Question 7: Discussion and Conclusions

The discussion was thorough and the conclusion was reasonable.

Response: I really appreciate your comments.

Reviewer 2 Report

The Authors made research regarding L-Tryptophan-dependent auxin producing plant growth promoting bacteria improve seed yield and quality of carrot by altering the umbel order. Comprehensive Results part. Please see my comments bellow:

L1. Type of the paper must be specified as Article. Remove all the other types of paper.

L38. Check the first sentence. Why not onions? Or insert reference supporting the statement regarding the carrot.

L97-100. Aim of the study is 1.5 lines. It must be presented separately in the last paragraph of the Introduction and needs to be addressed from the perspective of describing the contribution to the field under analysis and the elements of scientific novelty presented. Develop it better. Make the reader to understand the importance of your research.

L104. A satellite photo of the place/site would be relevant.

L184. Unbold the formula. Write it using Equation (check the Instructions for authors). Explain CFU abbreviation under it. Acronyms/Abbreviations/Initialisms should be defined the first time they appear in each of three sections: the abstract; the main text; the first figure or table. When defined for the first time, the acronym/abbreviation/initialism should be added in parentheses after the written-out form.

Subsection 2.8. All computer programs/softs used, and their variants must be mentioned and referenced. Please check and complete.

Table 1 should be inserted after the paragraph, not in the middle of it. Do the same for all figures/tables. They must be as closest is possible to the place they are mentioned in the text, but after finalizing that paragraph.

Under Table 1 explain the meaning of the letters a to d (same for ALL the Tables), and all the abbreviations used in the Table, according to the Instructions for authors. Same (regarding abbreviations) under each figure/table.

L425-428. Justified.

Discussion section:

How it can be improved the carrot crops management under the effects of climate change, a factor which has a huge importance nowadays? I suggest checking and referring to https://doi.org/10.1007/s11356-021-14127-7

Detail better the soil enzymes/ enzymology (https://doi.org/10.37358/RC.18.10.6590 and https://doi.org/10.37358/RC.17.10.5864 )

There is nothing mentioned about nanotechnology in agriculture, as it is one of the most modern ways in approaching your topic. Please complete, after checking https://doi.org/10.1016/j.chemosphere.2021.132533

After L521, as the last paragraph of Discussion section, please add the strengths and the weakness of your research.

Good English

Author Response

Response to Reviewer comments 2

We appreciate you for your precious time in reviewing our paper and providing valuable comments. It was your valuable and insightful comments that led to possible improvements in the current version. The authors have carefully considered the comments and tried our best to address every one of them. The comments and suggestions are valuable and very helpful for revising and improving our manuscript. We have made revisions according to the referees' comments and suggestions, as described in the authors' response. We would like to express our great appreciation to you for comments on our paper. Looking forward to hearing from you.

Question 1: L1. Type of the paper must be specified as Article. Remove all the other types of paper.

Response: Paper has been specified as Article. All other types of paper have been removed.

Question 2: L38. Check the first sentence. Why not onions? Or insert reference supporting the statement regarding the carrot.

Response: We have tried to modify the first sentence in line 38. Carrot roots are packed with Vitamin A, which is good for eye health. Carrots are particularly rich in carotene (pro-vitamin A). They are consumed either fresh, as a salad crop, or cooked. These roots also lower the risk of heart diseases. Therefore, include this vegetable in your diet to reap the myriad health benefits (Que et al., 2019).

Question 3: L97-100. Aim of the study is 1.5 lines. It must be presented separately in the last paragraph of the Introduction and needs to be addressed from the perspective of describing the contribution to the field under analysis and the elements of scientific novelty presented. Develop it better. Make the reader to understand the importance of your research.

Response: The aim of study was made clear by rewriting it in a more precise way. It is now presented separately in the last paragraph of the Introduction. Scientific novelty is also presented to understand the importance of the research.

Question 4: L104. A satellite photo of the place/site would be relevant.

Response: Unfortunately, we don’t have satellite photos of the site. Please accept our apologies.

L192-193. Unbold the formula. Write it using Equation (check the Instructions for authors). Explain CFU abbreviation under it.

Response: In L192-193 we have unbold the formula. Used equation form and CFU abbreviation has been added.

L194-196. Why only the Measurement of auxin production by PsJN were described?

Response: In L194-196 we have revised the title of heading 2.7 and text accordingly.

L1431-432. It should be also indicated that it was measured the dry weight of bacterial isolates.

Response: In L431. we have added in title and also in table.

Acronyms/Abbreviations/Initialisms should be defined the first time they appear in each of three sections: the abstract; the main text; the first figure or table. When defined for the first time, the acronym/abbreviation/initialism should be added in parentheses after the written-out form.

Response: Acronyms/Abbreviations/Initialisms have been defined the first time in each of three sections: the abstract; the main text; the first figure or table. The acronym/abbreviation/initialism has been added in parentheses after the written-out form.

Question 5: Subsection 2.8. All computer programs/softs used, and their variants must be mentioned and referenced. Please check and complete.

Response: All computer programs/softs used have been mentioned and referenced.

Table 1 should be inserted after the paragraph, not in the middle of it. Do the same for all figures/tables. They must be as closest is possible to the place they are mentioned in the text, but after finalizing that paragraph.

Response: In all tables we have added all abbreviations meanings in notes and same in case of graphs, all abbreviations meanings have been added.

Under Table 1 explain the meaning of the letters a to d (same for ALL the Tables), and all the abbreviations used in the Table, according to the Instructions for authors. Same (regarding abbreviations) under each figure/table.

Response: All tables have been added after the paragraph and they have been added as closest as possible to the place they are mentioned in the text. In all tables meaning of lettering has been mentioned.

L425-428. Justified.

Response: L425-428 has been Justified.

Question 6: Discussion section

How it can be improved the carrot crops management under the effects of climate change, a factor which has a huge importance nowadays? I suggest checking and referring to https://doi.org/10.1007/s11356-021-14127-7

Response: We agree with the honorable reviewer, however the study is related to umbel order alteration, so it is not possible to discuss related to climate change in our study.

Detail better the soil enzymes/ enzymology (https://doi.org/10.37358/RC.18.10.6590 and https://doi.org/10.37358/RC.17.10.5864 )

Response: We have studied enzymes (CAT, POD) in seeds. The reference has been cited in the revised manuscript.

There is nothing mentioned about nanotechnology in agriculture, as it is one of the most modern ways in approaching your topic. Please complete, after checking https://doi.org/10.1016/j.chemosphere.2021.132533

Response: We are agreed with honorable reviewer suggestion. However, our main focuss was to observe the effect of PGPB on growth and umbel seed quality of carrot. We shall consider the hionotable suggestion in our future experimentation.

After L521, as the last paragraph of Discussion section, please add the strengths and the weakness of your research.

Response: After L521, as the last paragraph of discussion section we have added the strengths and weaknesses of our research.

Reviewer 3 Report

The paper titled " L-Tryptophan-dependent auxin producing plant growth promoting bacteria improve seed yield and quality of carrot by altering the umbel order" describes the findings of a study using the exogenous application of L-Tryptophan-dependent auxin produced by plant growth promoting bacteria to improve carrot seed quality, growth indices, yield and related traits.

The study issue is interesting and pertinent in the field of agronomy. The paper was enjoyable to read. The title's meaning is unclear at first glance, but it may be changed. 

The introduction provides appropriate context for the study. The research design is appropriate, and the methods and results are well-described, although there are some minor suggestions/corrections that are indicated above.

Abstract and Introduction:

The purpose of the study, as well as the methodology used, should be made clearer, namely that the auxin precursor (L-TRP) and/or the bacterial isolates (MN54, MN17, MN34 and PsJN) were both used for foliar application. The study period (2-years) should also be mentioned.

The major purpose of study was to improve carrot seed quality; however, it should be noted that it also evaluated growth indices, yield, and related traits.

Results:

There are too many tables, making the reading difficult to follow. Some of the tables could possibly be moved to supplementary material.

It would be interesting (if possible) to include a figure showing the images of carrot umbel under the different treatments, with and without auxin precursor L-TRP

Conclusions:

Since the main purpose of study was to explore the potential of bacterial strains for altering umbel order to produce quality carrot seeds, it seems adequate to indicate in conclusions, besides de mention in abstract, to the bacterial strains that, according to the results, better suit the purpose

References: Try to update some of the references

There are some minor correction/suggestions in the attached pdf

Author Response

Response to reviewers comments 3

We are grateful for insightful comments on our paper entitled “L-Tryptophan-dependent auxin producing plant growth promoting bacteria improve seed yield and quality of carrot by altering the umbel order”. We have been able to incorporate changes to reflect most of the suggestions provided by the reviewer. We have highlighted the changes within the manuscript. Here is a point-by-point response to the reviewers' comments and concerns.

We would like to express our great appreciation to you for comments on our paper. Looking forward to hearing from you.

Question1: Abstract and Introduction
The purpose of the study, as well as the methodology used, should be made clearer, namely that the auxin precursor (L-TRP) and/or the bacterial isolates (MN54, MN17, MN34 and PsJN) were both used for foliar application. The study period (2-years) should also be mentioned.

The major purpose of study was to improve carrot seed quality; however, it should be noted that it also evaluated growth indices, yield, and related traits.

Response:

The aim of the study was made clear by rewriting it in a more precise way.

In methodology it was mentioned in Line “133-135” that both of them were used for foliar application. The study period for 2 years has been mentioned in line 106.

The main purpose of study was to improve the carrot seed quality because carrot seeds are produced in different types of umbels (particularly tertiary order umbels) and due to uneven maturation of carrot seeds produced in different umbel orders seed quality is variable.

Therefore, the objective of study was to suppress or reduce the number of tertiary umbels by application of “auxin producing bacterial strains”. It was also evaluated that besides improving seed quality, application of bacterial strains also enhanced the seed yield and growth-related attributes.

Question 2: Result
There are too many tables, making the reading difficult to follow. Some of the tables could possibly be moved to supplementary material.

Response: We have tried to compile them for easy understanding. These all tables are interlinked to different umbel orders and relevant to growth, yield and seed quality attributes so it is not possible to move them as supplementary material.

It would be interesting (if possible) to include a figure showing the images of carrot umbel under the different treatments, with and without auxin precursor L-TRP

Response: The images of carrot umbel under different treatments, for two years with and without auxin precursor L-TRP have been added as supplementary Figures 1-2).

Question 3: Conclusions

Since the main purpose of study was to explore the potential of bacterial strains for altering umbel order to produce quality carrot seeds, it seems adequate to indicate in conclusions, besides de mention in abstract, to the bacterial strains that, according to the results, better suit the purpose

Response: Thank you for your kind words. We have mentioned the selected bacterial strain (MN34+L-tryptophan) in abstract and conclusion for clarity, as potential strain to reduce number of tertiary umbels.

Question 4: References: Try to update some of the references

Response: We have tried to update the references.

Question 5: There are some minor correction/suggestions in the attached pdf

Response: Minor corrections/suggestions in the attached pdf have been incorporated and have highlighted the changes within manuscript.
